# Research on Roundness Error Evaluation of Connecting Rod Journal in Crankshaft Journal Synchronous Measurement

Tingting Gu [1,2], Xiaoming Qian [1,*] and Peihuang Lou [1,*]

1    College of Mechanical and Electrical Engineering, Nanjing University of Aeronautics and Astronautics, 29 Yudao Street, Qinhuai District, Nanjing 210016, China; gutingting@nuaa.edu.cn
2    Jincheng College, Nanjing University of Aeronautics and Astronautics, 88 Hangjin Avenue, Jiangning District, Nanjing 211156, China
*    Correspondence: drqian@nuaa.edu.cn (X.Q.); meephlou@nuaa.edu.cn (P.L.)

**Featured Application: To solve the problem that the measurement data of connecting rod journal is not consistent with the measurement angle in the synchronous measurement of the crankshaft journal, improve the measurement accuracy.**

**Abstract:** The crankshaft is the core part of an automobile engine, and the accuracy requirements of various shape and position errors are very high. On the basis of a synchronous measurement system, the connecting rod journal is deeply studied, including data processing and roundness evaluation. Firstly, according to the measuring processes of connecting rod journals, the real sampling angle distribution function was established, and the corresponding Gaussian weight function of each sampling angle was calculated. The weight function and the collected data corresponding to the angle were subjected to discrete cyclic convolution operation in the spatial domain to obtain the filtered effective circular contour data. Secondly, the particle swarm optimization algorithm was improved, and its inertia weight was set to decrease nonlinearly to speed up the convergence. A calculation process suitable for the evaluation of journal errors was designed. Then, the improved particle swarm optimization algorithm was used to evaluate the roundness of the corrected rod journal contour data. At last, through multiple measurement experiments, the feasibility and effectiveness of the synchronous measurement scheme and data processing method proposed in this paper are verified.

**Keywords:** synchronous measurement; connecting rod journal; non-equal interval sampling; particle swarm optimization; roundness error evaluation

## 1. Introduction

As the core component of an automobile engine, the diameter, roundness and cylindricity of crankshaft journals and other dimensional and shape errors directly affect the matching precision, running noise and vibration of engine parts. Therefore, it is very important to detect the error of crankshaft journals. At present, most of the crankshaft manufacturers in China have gradually realized the automation of crankshaft production, the production cycle has been shortened, and the precision has been greatly improved [1–3]. The improvement in detection technology is also imminent.

The research on detection technology of the automobile engine crankshaft abroad began relatively early. Since the 1960s, research on and manufacturing of crankshaft-measuring machines have been carried out. Up to now, there have been many detection methods of high measuring accuracy and perfect detection equipment by companies such as Hommel and Dr. Heinrich Schneider of Germany, Marposs of Italy, and Adcole of the United States [4–7]. The contact typed crankshaft comprehensive measuring equipment developed by Hommel Company of Germany is used in the final inspection of the crankshaft, which can complete the graded marking and automatic sorting of the diameter of the

main journal and the connecting rod journal [8,9]. The SKM series crankshaft measurement system developed by the company of Dr. Heinrich Schneider adopts a composite measurement method, which can, respectively, accomplish the contact and non-contact measurement combined with the contact displacement sensor and the area array charge-coupled device (CCD). According to the different characteristics of the measured object, the appropriate measurement method is chosen to obtain more measurement data and improve the measurement accuracy [10]. The products M2016 and M110 developed by the Italian company, Marposs, use the crankshaft measurement method based on the inductive probe, which can realize the automatic feeding, positioning and rotation of the workpiece. In addition to the measurement function, the product also has parts sorting, non-destructive testing and other functions. Another Optoquickset crankshaft measuring machine developed by the company uses linear array CCD to collect the contour characteristics of the crankshaft and installs multiple ultra-high-resolution linear array CCD [11–14]. The Adcole 1200 cylindrical coordinate measuring machine (CCMM) developed by Adcole is a high-precision measuring instrument for measuring camshafts, crankshafts and other cylindrical parts with strict tolerances. The equipment adopts advanced laser interferometer measurement technology and has a laser measurement reference system with high flatness, which provides accurate radial and length measurements [15].

In China, the crankshaft automatic detection technology started relatively late and, on the whole, is still in the exploratory stage. An online tracking measurement method was proposed for the non-circular grinding of a crankshaft connecting rod journal. However, the measurement accuracy was poor due to the influence of grinding debris and metal debris during the measurement process [16]. The V-shaped reference block was used to realize the quasi-online measurement of the roundness of the connecting rod journal. The contour data of the whole circle were obtained by equal interval sampling based on the scanning angle of the probe. The roundness error was separated out after signal processing, and the data were corrected [17,18]. An automatic precision measuring instrument for crankshaft groove based on backlight image was developed by Shanghai Jiaotong University. The optical system composed of a CCD camera, telecentric lens and backlight parallel light source was driven by a servo precision control system to detect the key feature dimensions of the crankshaft groove. Through the improvement of the image contour extraction algorithm and the error compensation for the mechanical system, the measurement accuracy, reliability and stability were improved. This backlight image measurement method had higher requirements for the measurement environment and a larger workload [19–21]. The method of follow-up measurement for non-symmetrical axis parts was designed. The hardware system of the automatic integrated measuring machine was studied, and the position relation model between the workpiece and measuring coordinate system was established. The comprehensive measurement software for shaft parts was also developed [22]. In view of the problem that the measurement accuracy decreases due to the width of the Abbe probe, the measurement method of a crankshaft was studied, and a crankshaft servo control method based on a four-axis motion system was proposed [23]. A comprehensive parameter measuring instrument for engine crankshafts was designed. The composition scheme, measurement principle and operation process of the measurement system were introduced. The least-squares method was adopted to process the data to reduce the measurement error [24].

The roundness and cylindricity of journals are important parameters of the crankshaft, which will directly affect the life and performance of the crankshaft. Because of the characteristics of the journal form and position error, scholars pay more attention to the study of the roundness error and use it as the basis of the evaluation of the cylindricity error. Many scholars in China and abroad have studied the evaluation methods of roundness error of components. Chetwyhd [25] studied the relationship between the reference worm and the circle and established the error calculation model of the minimum zone circle, the maximum inscribed circle and the minimum circumscribed circle. The results showed that it is convenient for computer processing to use reference worms instead of reference circles,

the precision was high enough under actual measurement conditions. Chiabert [26] evaluated the roundness error and its uncertainty of a workpiece by comparing the probability and statistics method with the classical LSM based on the global positioning system (GPS) standard. Srinivasu [27] proposed a hybrid method based on LSM and a new probabilistic global search Lausanne technique, which can accurately evaluate the roundness error of a workpiece. Luo [28] proposed an improved artificial bee colony algorithm to evaluate the roundness error. The roundness error evaluation example also proved that the algorithm had strong accuracy and stability in evaluating the roundness error. Cui [29] improved the particle swarm optimization algorithm to evaluate the roundness error. The algorithm conformed to the principle of the minimum zone method and had the advantages of strong global search ability, good stability and fast convergence speed. It was verified by an example that the accuracy of the algorithm for evaluating the roundness error is better. Du [30] proposed a novel PSO algorithm to evaluate the minimum zone roundness error by changing the inertia weight value and attaining its optimal value. The linear variable inertia weight PSO results were compared with the PSO with three different inertia weight values. Compared with GA and LSM, PSO based on inertia weight had advantages in evaluating accurate roundness error. Kumar [31] proposed a new teaching-learning-based optimization (TLBO) algorithm with fewer parameters for roundness error measurement, and the results were compared with the classical PSO algorithm. By comparing the accuracy and convergence time of the two advanced optimization algorithms, it was found that the two algorithms have obtained similar results, while TLBO takes a higher calculation time in comparison to PSO. Pathak [32] applied the constriction factor PSO (CFPSO) algorithm for evaluating different form errors, including minimum zone roundness. The constriction factor was added to the group velocity updating equation to enhance the exploration in initial iterations. The CFPSO algorithm provided improvement in roundness error when compared with LSM, GA and other algorithms and also had a faster convergence rate. At present, the traditional error evaluation algorithm and the intelligent error evaluation algorithm are mainly used to evaluate the profile error of the simple shape. The roundness error evaluation is essentially a nonlinear optimization problem, it is difficult to solve with traditional methods, but it is more convenient to solve nonlinear optimization problems by intelligent algorithms. There are many intelligent algorithms used in roundness error evaluation.

To sum up, crankshaft detection equipment technology's maturity in China is low, and there is still a gap compared with foreign advanced technology and equipment. Based on the synchronous measurement method of the contact crankshaft journal, the measurement of the connecting rod journal circular profile is further studied. Firstly, in the process of the synchronous measurement of the journal, the sampling angle of the connecting rod journal has the problem of a non-uniform interval. The weight function of each sampling angle is calculated according to the actual distribution of sampling angles. Gauss filter is used to obtain the modified circular profile to solve the problem of unequal interval sample data processing. Then, the particle swarm optimization algorithm is improved, and the inertia weight is set as nonlinear, decreasing to accelerate the convergence. Finally, the roundness error of the corrected measurement data is evaluated by an improved particle swarm optimization algorithm.

## 2. Synchronous Measurement Structure of Crankshaft Journal

### 2.1. Probe Arrangement

The synchronous measurement scheme for the crankshaft journal is to measure the spindle journal and connecting rod journal synchronously under one-time clamping. Multiple probes are arranged in both axial and radial directions of the journal to shorten the measurement time and improve the measurement accuracy. When the probe is in contact with the circumferential surface of the main journal and the connecting rod journal, one revolution of the crankshaft can not only measure the surface profile data of the main journal but also collect the whole surface profile data of the connecting rod journal. The

mechanical structure of multi-probe is relatively simple, the adaptability to the environment is high, and the anti-interference ability is strong. It can eliminate machining errors, as well as installation and clamping errors, and can be used for error separation.

The profile for the measured crankshaft and the distribution of measured points is shown in Figure 1. Among them, J1–J5 are the main journals, and P1–P4 are the connecting rod journals. In the measurement process, the spindle journals J1–J5 perform autorotation motion around the rotation axis, and the connecting rod journals P1–P4 perform circular motion around the rotation axis. The diameter, roundness, cylindricity, and concentricity of each journal of the crankshaft can be calculated by one rotary motion. This method realizes the synchronous measurement of crankshaft multi-journal and multi-parameters.

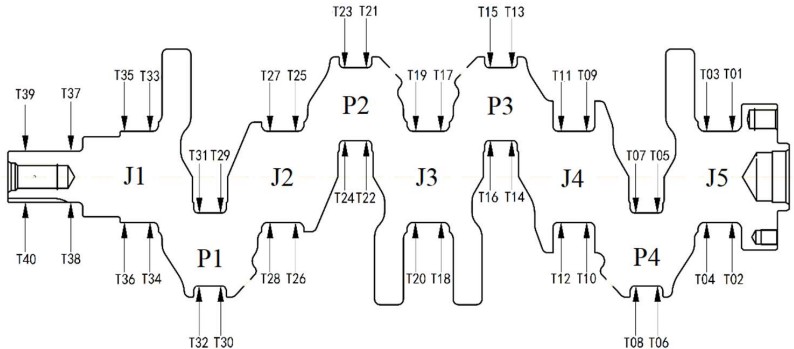

**Figure 1.** Section and probe arrangement of crankshaft.

### 2.2. Realization of Synchronous Measurement of Connecting Rod Journal

As shown in Figure 2, the main shaft journal arm and the connecting rod journal arm are fixed on a lifting–lowering pipe. After the crankshaft is clamped, the boom is driven down by the lifting–lowering pipe to carry out the synchronous measurement of all the journal bearings. During the measurement process, the measuring arm of the connecting rod journal simulates the motion state of the connecting rod and follows the movement away from the connecting rod journal. At the same time, the measuring arm follows the connecting rod journal to simulate the motion state of the connecting rod. The radial data of a complete circle of the surface of the connecting rod journal can be acquired while the connecting rod journal rotates around the main journal.

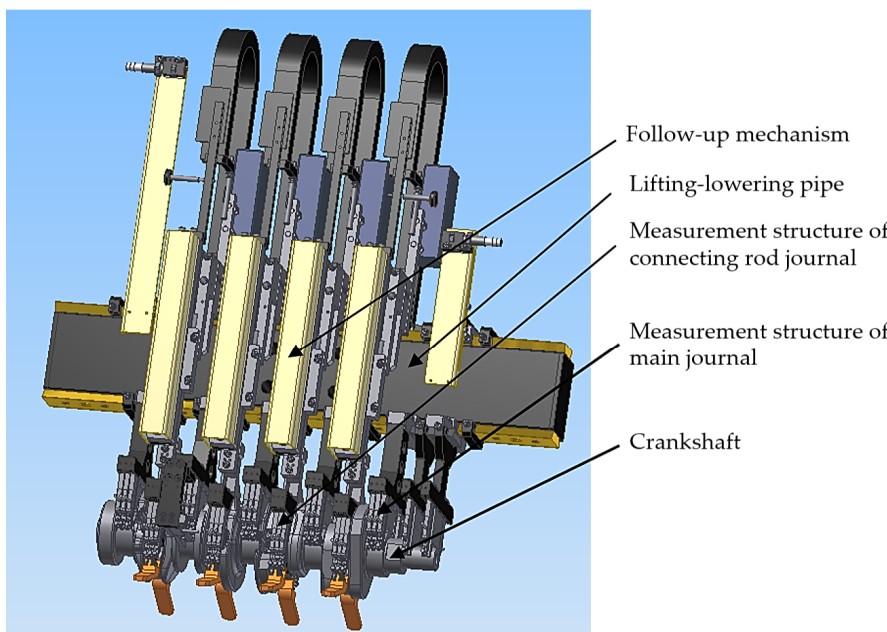

**Figure 2.** Synchronous combined measurement arm.

### 3. Measurement of Connecting Rod Journal

*3.1. Relationship between Sampling Angle and Motion Angle*

To realize the synchronous measurement of the crankshaft main journal and connecting rod journal, the measuring mechanism can follow the movement away from the connecting rod journal to complete the cross-section radial data collection without interfering with the measuring structure of the main journal. To simulate the motion state of the connecting rod when the engine is running, the crank arm of the crankshaft is regarded as a crank, the connecting rod journal and the measuring head part are regarded as a hinge and the measuring arm is regarded as a connecting rod. A crank-rocker measuring mechanism is designed, and the principle of the mechanism is shown in Figure 3. $O$ is the center of the main journal, $O_1$ is the center of the connecting rod journal, $OO_1$ is the crank arm, $O_1B$ is the measuring arm. Probe $A$ is installed in the direction perpendicular to the measuring arm, and guide rail $B$ is fixed on the lifting–lowering pipe. When guide rail $B$ descends to the measuring point, the distance from the center of the main journal is $L$. The measuring arm $O_1B$ is a retractable structure, which can expand and contract with the rotary motion of the connecting rod journal.

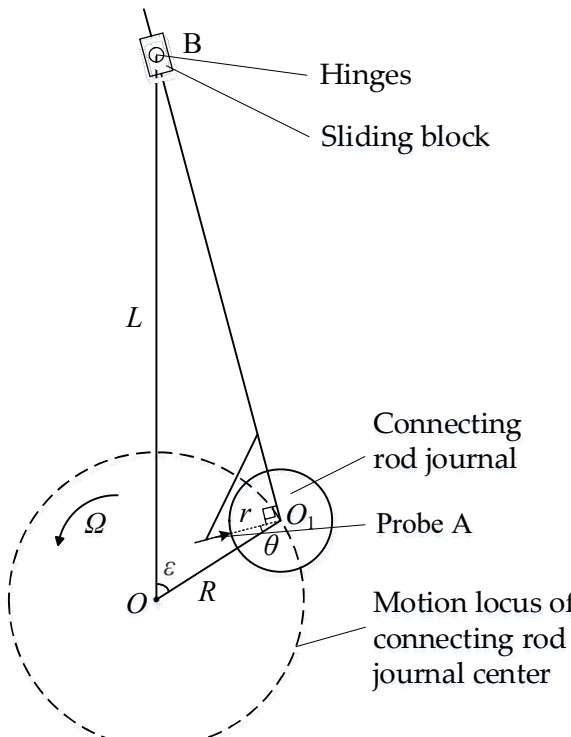

**Figure 3.** Principle diagram of connecting rod journal measurement mechanism.

According to the schematic diagram of the measuring mechanism shown in Figure 3, the center $O$ of the circle is the center of rotation, and the circular dotted line is the trajectory of the connecting rod journal center $O_1$. The radius $R$ of the dotted circle is the length of the crank arm, the radius $r$ of the realization circle is the radius of the connecting rod journal. The angle between the crank arm $OO_1$ and the vertical lifting table $OB$ is $\varepsilon$. In the process of crankshaft measurement, the counterclockwise direction is set as positive, and the angular velocity is $\omega$. The angle between probe $A$ in contact with the connecting rod journal and the crank arm $OO_1$ is set as $\theta$. According to the cosine theorem of $\Delta OO_1B$, the functional relationship between the included angle $\theta$ and the rotation angle $\varepsilon$ is:

$$\theta = \begin{cases} \frac{\pi}{2} - \arccos\left(\frac{R - L\cos\varepsilon}{\sqrt{R^2 + L^2 - 2RL\cos\varepsilon}}\right), 0° \leq \varepsilon \leq 180° \\ \frac{\pi}{2} + \arccos\left(\frac{R - L\cos\varepsilon}{\sqrt{R^2 + L^2 - 2RL\cos\varepsilon}}\right), 180° < \varepsilon \leq 360° \end{cases} \tag{1}$$

According to Equation (1), the relationship curve between $\theta$ and $\varepsilon$ can be obtained, as shown in Figure 4. When the journal rotates once a week, the angle $\theta$ relative to the crank arm varies from $-90°$ to $270°$ as angle $\varepsilon$ varies from $0°$ to $360°$. The probe scans the circumferential surface of the connecting rod journal and collects the radial dimension change while the main journal rotates around the rotary center. However, it is found that the relationship between the angle $\theta$ and rotation angle $\varepsilon$ is not linear. The relationship between the angular velocity $\Omega$ and the angular $\varepsilon$ is obtained by the derivation of the included angle $\theta$, as shown in Figure 5. It is found that the motion of the probe on the circumferential surface of the connecting rod journal is not uniform, and the sampling interval is not equal.

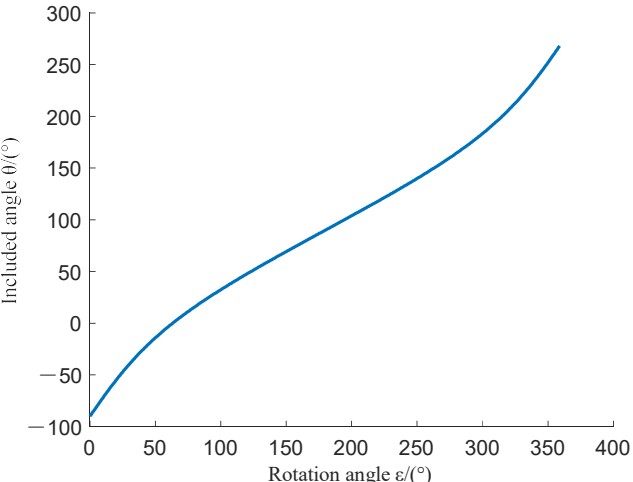

**Figure 4.** Relationship between included $\theta$ and rotation angle $\varepsilon$.

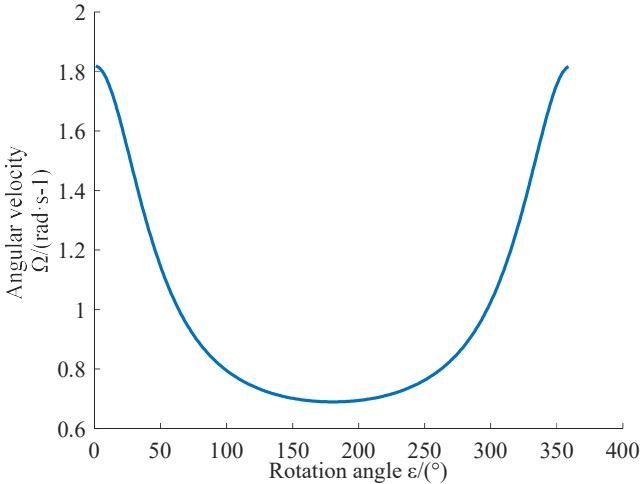

**Figure 5.** Relationship between angular velocity $\Omega$ and rotation angle $\varepsilon$.

### 3.2. Data Processing

According to Equation (1) and the relation curve, it is found that the contour data collected by the connecting rod journal probe are non-equal interval samples. A method was proposed to multiply the amplitude of the original signal by the interval time between two signals, then take the Fourier transform [33]. In fact, the samples collected will change with time and should be processed first. The method based on the assumption of equal interval samples and Fourier transform in the frequency domain is no longer suitable. It is necessary to operate according to the actual distribution of the sampling angles of each sample and then deal with it by discrete Gauss weighting function.

In ISO 16610 standard, the weight function of the Gaussian low-pass filter for workpiece surface texture analysis can be given by Equation (2):

$$s(x) = \frac{1}{\alpha \lambda_c} \exp\left[-\pi\left(\frac{x}{\alpha \lambda_c}\right)^2\right] \tag{2}$$

where $x$ is the distance from the center of symmetry of the weight function, $\lambda_c$ is the cutoff wavelength of Gaussian low-pass filter and $\alpha$ is a constant.

In the filtering process of the circular profile with a cross-section, the cutoff wavelength $\lambda_c$ is usually replaced by the cutoff wave number $N_C$. In this scheme, the measured journal is compared with the calibration part, and the sampling data are the variations in diameter of all journals relative to the calibration part. The weight function of the Gaussian low-pass filter does not depend on the diameter of the journals, so Equation (2) can be expressed in cutoff wave number $N_C$. The weight function can be discretized in the case of equal interval sampling and is expressed as:

$$s(p\Delta\theta) = \frac{N_c}{\alpha} \exp\left[-\pi\left(\frac{p\Delta\theta N_c}{\alpha}\right)^2\right] \tag{3}$$

where $\Delta\theta$ is the sampling interval angle, and $p$ represents the $p$-th sampling angle. In fact, the sampling interval angle $\Delta\theta$ of connecting rod journal varies non-uniformly, which is related to the distance $L$ of the measuring point and the length $R$ of the crank arm. Compared with the equal interval sampling, the weight function of the Gauss filter for non-equally samples is determined by the specific sampling angle in Equation (1). The corresponding weight function is different, while the actual sampling angle is different. Therefore, the weight function for each sampling angle can be obtained, which is expressed as follows.

$$s(p-q) = \frac{N_c}{\alpha} \exp\left[-\pi\left(\frac{(\theta_p - \theta_q)N_c}{\alpha}\right)^2\right]$$
$$(p, q = 1, 2, \ldots, M) \tag{4}$$

$\theta_p$, $\theta_q$ represent different sampling angles, and $M$ represents the number of samples taken by a single probe around the connecting rod journal. By normalizing, Equations (4) and (5) can be obtained.

$$\widetilde{s}(p-q) = \frac{s(p-q)}{\sum_{p=1}^{M} s(p-q)}, p, q = 1, 2, \ldots, M \tag{5}$$

The angle information is used to calculate the coefficients of the filter and the amplitude information is multiplied by the angle information. The discrete cyclic convolution sum is directly used for Gaussian filtering through the spatial domain, and the non-equidistant Gauss filtered data are:

$$y(p) = \sum_{q=1}^{M} \widetilde{s}(p-q)\rho(q), p, q = 1, 2, \ldots, M \tag{6}$$

where $\rho(q)$ is the original contour data, and $y(p)$ is the data filtered by unequal interval weight function.

## 4. Roundness Error Evaluation

### 4.1. Mathematical Model

The most commonly used methods for roundness error evaluation are the least-squares method (LSM), the minimum zone circle (MZC) method, the maximum inscribed circle (MIC) method and the minimum circumscribed circle (MCC) method [34]. In this paper,

LSM and MZC are used to calculate the roundness error at the same time. The mathematical models of the two methods are described in Equations (7) and (8).

Let $Xi\ (x_i,\ y_i)$, $(I = 1,2,3, \dots, n, n > 3)$ be the coordinates of the measuring points in the actual circumference of the circle under test. The mathematical model of the least-squares method is:

$$f(a_L, b_L, R_L) = min \sum_{i=1}^{n} \left( \sqrt{(x_i - a_L)^2 + (y_i - b_L)^2} - R_L \right)^2 \qquad (7)$$

where $(a_L, b_L)$ and $R_L$ are the center and radius of the least-squares circle, respectively.

The mathematical model of the minimum zone method is:

$$f(a_M, b_M) = min \left( max \sqrt{(x_i - a_M)^2 + (y_i - b_M)^2} - min \sqrt{(x_i - a_M)^2 + (y_i - b_M)^2} \right) \qquad (8)$$

where $(a_M, b_M)$ is the center of the minimum zone circle.

### 4.2. Particle Swarm Algorithm

#### 4.2.1. Common Particle Swarm Algorithm

Particle swarm optimization (PSO) [35] is an optimization algorithm based on swarm intelligence theory, which is essentially a random search algorithm developed by simulating the foraging behavior of birds. In the foraging behavior of the birds, each individual's foraging behavior seems to be random, but the individuals that have found food play a certain role in guiding the whole foraging population. The individual's current foraging state preserves relevant experience for subsequent foraging behavior, and the individual in the birds judges that it is constantly moving to the area around the nearest individual to find the area where food is most likely to be present.

In the particle iteration process, let the position of the $I$ particle in the d-dimensional solution space be expressed as $X_i = (x_{i1}, x_{i2}, x_{i3} \cdots x_{id})$, and its velocity is $V_i = (v_{i1}, v_{i2}, v_{i3} \cdots v_{id})$. The particle updates the velocity and position of the individual particle according to the individual optimal value $p_i = (p_{i1}, p_{i2}, p_{i3} \cdots p_{id})$ and the global or local optimal value $p_g = \left( p_{g1}, p_{g2}, p_{g3} \cdots p_{gd} \right)$ through Equations (9) and (10):

$$v_{id}^{t+1} = \omega v_{id}^t + c_1 r_1 \left( p_{id}^t - x_{id}^t \right) + c_2 r_2 \left( p_{gd}^t - x_{gd}^t \right) \qquad (9)$$

$$x_{id}^{t+1} = x_{id}^t + v_{id}^{t+1} \qquad (10)$$

where $t$ is the number of iterations, $\omega$ is the inertia weight, $r_1$ and $r_2$ are two independent random numbers uniformly distributed in [0,1] and $c_1$ and $c_2$ are acceleration factors, which are generally constants.

#### 4.2.2. Setting of Inertia Weight

Inertia weight is an important parameter in the PSO algorithm. A larger weight is beneficial to improve the global search ability of the algorithm and avoid premature convergence caused by falling into a local minimum value. While smaller weight is beneficial to achieve precise search of the search area and improve the convergence accuracy. Although the linear decreasing weight can balance the global exploration ability and local search ability to a certain extent, it will make the local search ability worse in the later stage. The selection of inertia weight is directly related to the development ability and exploration of the algorithm. Through many experiments, the inertia weight is set to nonlinear decline, as shown in Equation (11).

$$\omega = \omega_{min} + (\omega_{max} - \omega_{min}) \sqrt{\frac{t}{N}} \qquad (11)$$

where $\omega_{max}$ is the maximum value of the inertia weight, $\omega_{min}$ is the minimum value of the inertia weight, $t$ is the current iteration number and $N$ is the maximum iteration number.

*4.3. Application of Particle Swarm Optimization in Error Evaluation*

PSO is widely used because of its easy implementation, high precision and fast convergence. The application process of particle swarm optimization in roundness error evaluation is as follows:

Step 1 Select particles

In the application of PSO to analyze the roundness error, the real number encoding method is adopted, and the particle form can be determined as follows:

$$X_i = (x_i, y_i)$$

Step 2 Initiate the particle swarm

According to the experimental results, set a reasonable particle swarm size $n$, set maximum velocity $V_{\max}$ and minimum velocity $V_{\min}$, population boundary $pop_{\max}$ and $pop_{\min}$, evolution times $T$, initial particle swarm $P(0)$ and initial velocity $V(0)$.

Since the least-squares method has a certain precision, the correlation parameters in the region near the center of the least-squares circle are set to the initial particle swarm.

$$P(0) = \left\{ \left( x_i^0, y_i^0 \right) \middle| i = 1 \cdots n \right\}$$

When particle swarm optimization is applied, the maximum velocity can be set to the desired precision according to its direction moving principle, and the expansion or divergence of particle swarm can be limited. The reasonable boundary of the population limits the control search scope and avoids the waste of computing resources. The inertia weight is reduced nonlinearly according to Equation (11).

Step 3 Calculate particle fitness and compare

In particle swarm optimization algorithm, the high fitness is regarded as a better case. The objective function of finding the minimum radius difference in the minimum region method is taken as a negative number, and the fitness function is set as:

$$fit(x, y) = r_{min} - r_{max}$$

The fitness values of all particles in the current population are calculated, the fitness values of all particles in the current population are compared with the historical fitness values of individual particles. Update the current global fitness extreme value $p_g = \left( p_{g1}, p_{g2}, p_{g3} \cdots p_{gd} \right)$ and update the individual particle fitness extreme value $p_i = (p_{i1}, p_{i2}, p_{i3} \cdots p_{id})$.

Step 4 Update speed and location

The velocity $v_i$ and position $x_i$ of each particle in the current particle swarm are updated according to Equations (9) and (10) to form a new particle swarm $P(t)$.

Step 5 Set stop conditions

When the iteration number $t$ does not reach the maximum number $T$ or the precision does not reach the set precision, it returns to calculate the fitness value of individual particles and updates the particles until the termination condition is reached and the results are output.

## 5. Experimental Verification and Discussion

*5.1. Experimental Platform*

The experimental platform is a crankshaft measuring instrument, the prototype is shown in Figure 6. The synchronous measuring module of the journal is installed on the same lifting base according to the number of measured journals. The height of the lifting–lowering pipe is controlled by a proximity switch and stops when it drops to the set height. At this time, the measuring arm of the measuring structure begins to descend until it reaches both sides of the crankshaft axis. The measuring module under each measuring structure integrates two groups of probes. Each set of probes is composed of two opposed

linear variable differential transformer (LVDT) sensors, which are perpendicular to the measuring arm. A laser locator is used to precisely position the rotary axis and reduce the error. The LVDT sensor probe is calibrated with a calibration element, and then the measured crankshaft is tested.

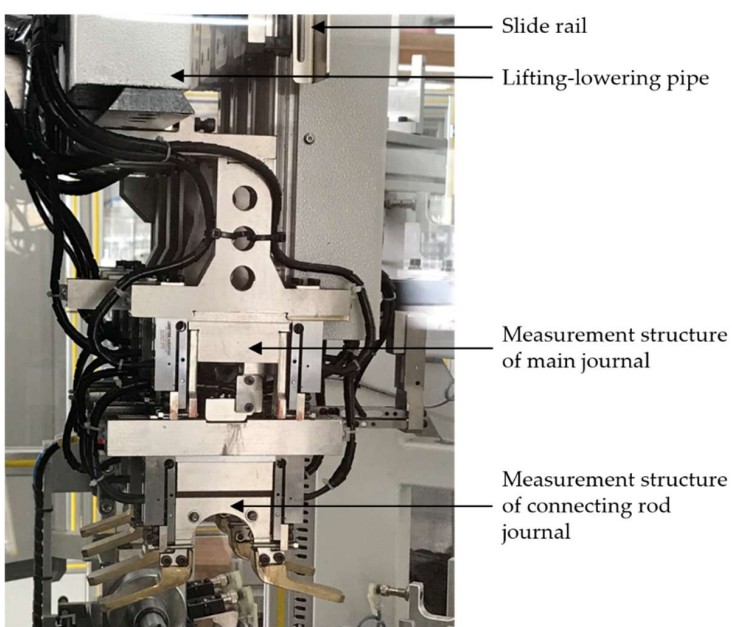

**Figure 6.** Experimental prototype of measuring crankshaft.

The measuring structure of the connecting rod journal and the distribution of probes are shown in Figure 7. The two follow-up wheels above the probes are always in contact with the surface of the crankshaft connecting rod journal and make the probe connect through the center of the connecting rod journal so as to ensure the probe collects the radial dimension change of the connecting rod journal during the rotation. Two guide bars are arranged under the probes to enable the connecting rod journal to enter the measuring point even if the initial state of the connecting rod journal deviates a little.

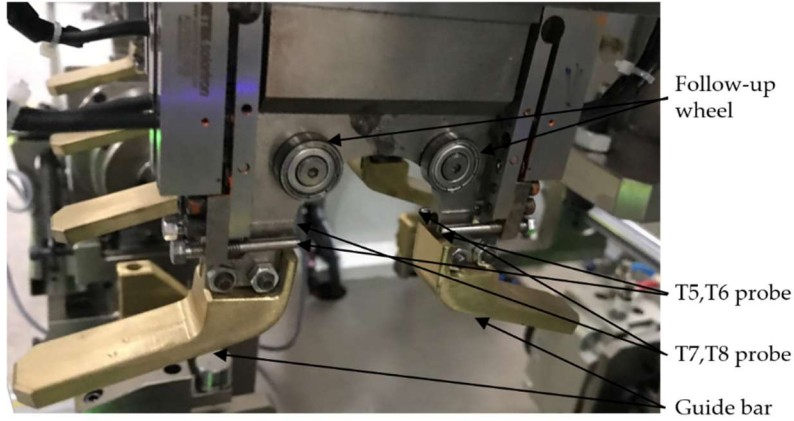

**Figure 7.** Measuring structure and probe arrangement of connecting rod journal.

### 5.2. Filter Measurement Data

The probe moved at a non-uniform speed on the circumference of the connecting rod journal, so the sampling data were arranged at non-equal intervals. According to Equation (1), the true sampling angle can be obtained. The circular profile of the connecting rod journal was obtained by the distribution of the sampling data corresponding to the

real sampling angle, which is shown in Figure 8. It can be found that with the rotation of the crankshaft, the sampling points on the connecting rod journal are first sparse and then dense, and the sampling interval is first long and then short. The corrected contour data were consistent with the real sampling angle distribution.

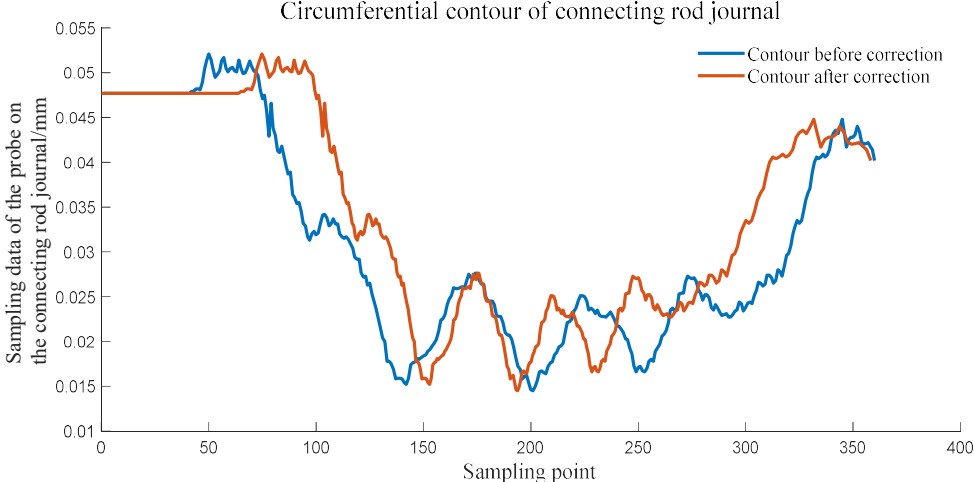

**Figure 8.** Radial sampling data of the connecting rod journal.

Based on the true distribution of the sampling angle and the data of the circumference of connecting rod journal measured by the sensor, the connecting rod journal P1, P2, P3 and P4 were filtered, respectively, by the method mentioned in this paper. The cut-off wave number was 50UPR, and the filtered contours of the four connecting rod journals can be obtained as shown in Figure 9a–d.

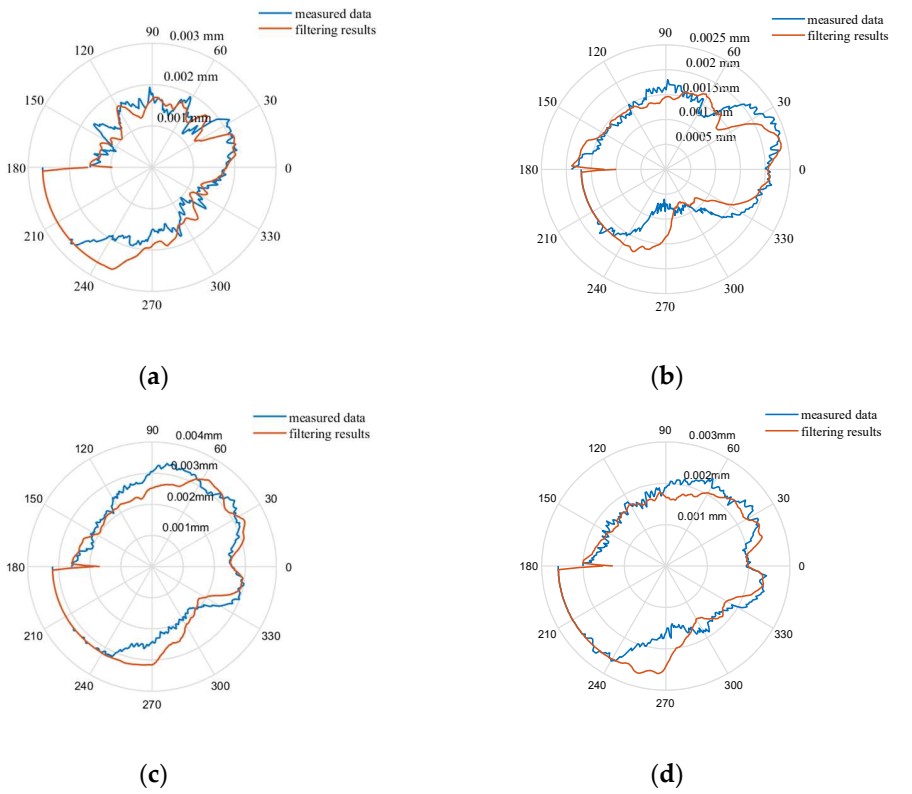

**Figure 9.** Filter profile of non-equal for all connecting rod journals: (**a**) The filtered contour of journal P1; (**b**) the filtered contour of journal P2; (**c**) the filtered contour of P3; (**d**) the filtered contour of P4.

### 5.3. Validation of Roundness Error Evaluation Method

The roundness error of the connecting rod journal was evaluated according to the filtered profile data. The initial particle swarm was selected as the random coordinates in the region around the $O_L$ according to the position of the center of the least squares. The parameters of PSO were selected as follows: particle swarm size $n$ = 50, iteration number $T$ = 100, learning factor $c_1 = c_2$ = 1.49, maximum inertia weight $\omega_{max}$ = 0.9 and minimum inertia weight $\omega_{min}$ = 0.4, maximum velocity $V_{max}$ = 0.001 and minimum velocity $V_{min}$ = −0.001, population boundary = [−0.2,0.2].

The genetic algorithm (GA), particle swarm optimization (PSO) and improved particle swarm optimization (IPSO) were used to calculate the roundness of crankshaft connecting rod journal P4. The convergence of the algorithms is shown in Figure 10.

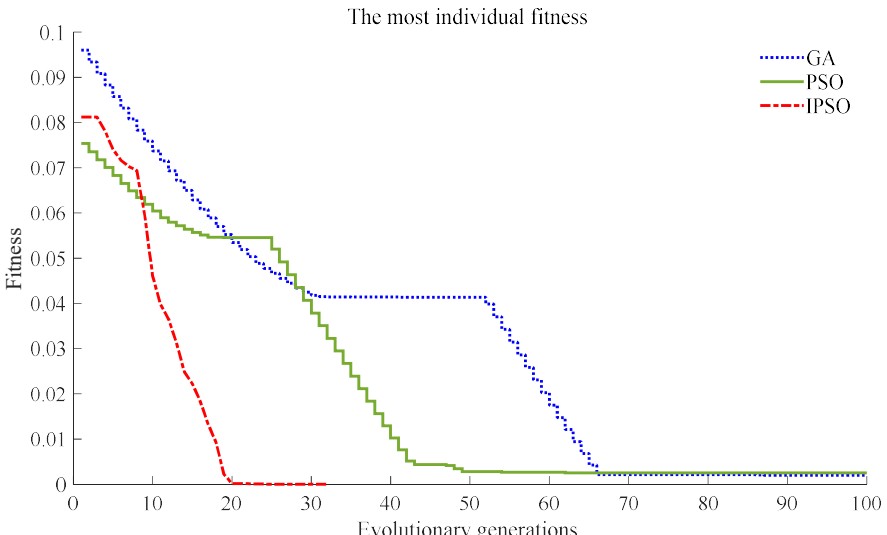

**Figure 10.** Convergence process of calculating P4 measuring section of the crankshaft sample.

The same method was used to perform five independent calculations on the measurement section, and the results are shown in Table 1.

**Table 1.** Results of 5 independent calculations on the measured data of crankshaft sample P4 using improved PSO.

| Number | Center of Circle ($10^{-3}$ mm) | Evolutionary Generations | Inscribed Point | External Point | Diameter/mm | Roundness/μm |
|---|---|---|---|---|---|---|
| 1 | (0.020, −0.049) | 21 | 242 | 383 | 39.9876 | 1.87 |
| 2 | (0.051, −0.016) | 19 | 438 | 383 | 39.9877 | 1.85 |
| 3 | (0.061, −0.016) | 32 | 438 | 383 | 39.9877 | 1.87 |
| 4 | (0.050, −0.026) | 27 | 438 | 383 | 39.9877 | 1.84 |
| 5 | (0.062, −0.057) | 31 | 438 | 383 | 39.9876 | 1.86 |

Improved particle swarm optimization (IPSO) was used to calculate the connecting rod journal P4 independently five times. The calculated positions of the center of the circle are close, and the deviation of the diameter is less than 0.1 μm. The general population of 20/30 generations can evolve into the optimal population, and the convergence speed is fast and stable. The results of various numerical calculations are basically the same, which shows that the particle swarm optimization algorithm has high accuracy.

Several evaluation methods were used to calculate the connecting rod journal P4 of the crankshaft sample, the results were compared, as shown in Table 2.

**Table 2.** The calculation results of each evaluation method of shape and position error.

| Method | Center of Circle ($10^{-3}$ mm) | Evolutionary Generations | Inscribed Point | External Point | Diameter/mm | Roundness/μm |
|--------|-------------------------------|-------------------------|-----------------|----------------|-------------|--------------|
| LSC | (0.107, 0.042) | - | - | - | 39.9877 | 2.23 |
| GA | (0.047, −0.028) | 65 | 242,438 | 383 | 39.9877 | 1.98 |
| IPSO | (0.049, −0.033) | 26 | 242,438 | 383 | 39.9877 | 1.86 |

Through comparison, it is found that the calculation error of diameter is small and the calculation error of roundness is large in the results of the least-squares method. It shows that the diameter value calculated by the least-squares method can ensure certain accuracy in the case of more sampling points, but there is still a certain error in the results of roundness.

By comparing the calculated results of two intelligent algorithms, it is found that the calculated center position error is small and all of them are around the center of the least square circle, which verifies the feasibility of the least-squares center as the starting point of the optimization algorithm. The positions of the minimum inscribed point and the maximum external point calculated by the genetic algorithm and particle swarm algorithm are basically the same, and the errors of diameter are small, which verifies the feasibility of the intelligent optimization algorithm. The global search performance of the genetic algorithm is better, while the local search performance of the particle swarm algorithm is better. After determining the optimal solution around the least-squares center, the inertia weight of the particle swarm algorithm is improved to find the optimal solution faster.

## 6. Conclusions

In this paper, the most critical journal measurement of the crankshaft measurement is studied, and a synchronous measurement scheme for all crankshaft journals is proposed to meet the measurement requirements of high repeatability and high efficiency.

The weight function corresponding to each sampling angle is calculated according to the actual distribution of the sampling angle. The weight function corresponding to each sampling angle and the sampling data corresponding to this angle are calculated by the time domain discrete convolution. The filtering weight function is obtained point by point through the angular velocity of the rotation axis and the sampling angle distribution obtained by the sampling interval time of the probe, which solves the problem of non-equal interval sample data processing.

The evaluation method of journal roundness is studied. The principle of particle swarm intelligence algorithm is improved, and a reasonable process suitable for roundness calculation is designed. Finally, the improved particle swarm evaluation method for roundness error is applied to specific experimental calculations and compared with the genetic algorithm. The feasibility of the intelligent algorithm in roundness error evaluation is verified, and the improved particle swarm algorithm converges faster than the common algorithm.

## 7. Patents

Processing method of crankshaft connecting rod neck data.
Application number: CN202110923239.6
Application date: 2021.08.12
Publication number: CN113704908A
Open date: 2021.11.26

**Author Contributions:** Conceptualization, T.G., P.L. and X.Q.; methodology, T.G. and X.Q.; software, T.G.; validation, T.G., X.Q. and P.L.; formal analysis, T.G. and X.Q.; investigation, X.Q.; resources, X.Q. and P.L.; data curation, T.G. and X.Q.; writing—original draft preparation, T.G.; writing—review and editing, T.G. and X.Q.; visualization, T.G.; supervision, X.Q.; project administration, P.L. and X.Q.; funding acquisition, X.Q. and P.L. All authors have read and agreed to the published version of the manuscript.

**Funding:** This research was funded by the Key R&D Project of Jiangsu Province, grant number BE20160034.

**Institutional Review Board Statement:** Not Applicable.

**Informed Consent Statement:** Not Applicable.

**Data Availability Statement:** Not Applicable.

**Conflicts of Interest:** The authors declare no conflict of interest.

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
