# Peer review of "Research on Roundness Error Evaluation of Connecting Rod Journal in Crankshaft Journal Synchronous Measurement"

_applsci, doi:10.3390/app12042214_

Round 1

Reviewer 1 Report

An excellent paper on roundness errors in crankshafts.

Author Response

Thanks for the reviewer's comments!

Reviewer 2 Report

Dear Authors,

I read the paper and from my point of view your work is well written. The paper contains all necessary chapters. An abstract is clear. Introduction contains necessary literature analysis without extensive self-citations. Experiment and data analysis do not raise any doubts. English language and style are good enough and from my point of view an additional revision is not necessary. I can recommend your work for publication in the present form. Congratulations.

Yours sincerely, Reviewer

Author Response

Thanks for the reviewer's comments!

Reviewer 3 Report

The present manuscript describes a research on roundness error evaluation of connecting rod journal in crankshaft journal synchronous measurement. Although the manuscript is well written, it lacks of the necessary novelty for a journal such as Applied Sciences. The methodology is too simple and the utility to the scientific community is not clear.

Author Response

Thanks for the reviewer's comments!

The crankshaft measurement machine can realize the synchronous measurement of all the journals of the crankshaft under one clamping. The measurement precision and repeatability are very high. There is almost no such study in China, which has been put into use. In terms of data processing, for the fast-paced production line, this method can not only ensure measurement accuracy but also keep up with the measurement speed to achieve full inspection.

Reviewer 4 Report

The presented paper is very well done. However, I have some recommendations:
- The introduction is too extensive and contains abbreviations that are not explained.
- unification of marking of parts in the text e.g. J1-J5 and in Figure 1J, resp. 5J, the same for "P"
- lines 320, 392-395 - font - unify
- line 397 - edit the character description according to the template
- pictures 6 and 7 - edit the lines on the labels - they are lost in the picture
- the literature contains few current sources
What will be your next research or implementation? 

Round 2

Reviewer 3 Report

I still thing that this manuscript lacks of the necessary novelty for a journal such as Applied Sciences. The methodology is too simple and the utility to the scientific community is not clear.